# ACE Phenotyping in Human Blood and Tissues: Revelation of ACE Outliers and Sex Differences in ACE Sialylation

**DOI:** 10.3390/biomedicines12050940

**Published:** 2024-04-23

**Authors:** Enikő E. Enyedi, Pavel A. Petukhov, Alexander J. Kozuch, Steven M. Dudek, Attila Toth, Miklós Fagyas, Sergei M. Danilov

**Affiliations:** 1Division of Clinical Physiology, Department of Cardiology, University of Debrecen, 22 Moricz Zs., 4032 Debrecen, Hungaryatitoth@med.unideb.hu (A.T.); 2Kálmán Laki Doctoral School of Biomedical and Clinical Sciences, University of Debrecen, 4032 Debrecen, Hungary; 3Department of Pharmaceutical Sciences, College of Pharmacy, University of Illinois at Chicago, 833 S. Wood Ave., Chicago, IL 60612, USA; pap4@uic.edu; 4Department of Medicine, Division of Pulmonary, Critical Care, Sleep and Allergy, University of Illinois at Chicago, CSB 915, MC 719, 840 S. Wood Ave., Chicago, IL 60612, USA; alexkozuch@gmail.com (A.J.K.); sdudek@uic.edu (S.M.D.)

**Keywords:** angiotensin I-converting enzyme, precision medicine, outliers, conformational changes, screening, albumin, CCL18

## Abstract

Angiotensin-converting enzyme (ACE) metabolizes a number of important peptides participating in blood pressure regulation and vascular remodeling. Elevated ACE expression in tissues (which is generally reflected by blood ACE levels) is associated with an increased risk of cardiovascular diseases. Elevated blood ACE is also a marker for granulomatous diseases. Decreased blood ACE activity is becoming a new risk factor for Alzheimer’s disease. We applied our novel approach—ACE phenotyping—to characterize pairs of tissues (lung, heart, lymph nodes) and serum ACE in 50 patients. ACE phenotyping includes (1) measurement of ACE activity with two substrates (ZPHL and HHL); (2) calculation of the ratio of hydrolysis of these substrates (ZPHL/HHL ratio); (3) determination of ACE immunoreactive protein levels using mAbs to ACE; and (4) ACE conformation with a set of mAbs to ACE. The ACE phenotyping approach in screening format with special attention to outliers, combined with analysis of sequencing data, allowed us to identify patient with a unique ACE phenotype related to decreased ability of inhibition of ACE activity by albumin, likely due to competition with high CCL18 in this patient for binding to ACE. We also confirmed recently discovered gender differences in sialylation of some glycosylation sites of ACE. ACE phenotyping is a promising new approach for the identification of ACE phenotype outliers with potential clinical significance, making it useful for screening in a personalized medicine approach.

## 1. Introduction

Angiotensin I-converting enzyme (ACE, CD143) is a Zn^2+^ carboxydipeptidase that plays key roles in the regulation of blood pressure and in the development of vascular pathology. ACE is constitutively expressed on the surface of endothelial cells, absorptive epithelial and neuroepithelial cells, and cells of the immune system (macrophages, dendritic cells)—reviewed in [1,2]. Circulating blood ACE originates from endothelial cells [3], primarily lung capillary endothelium [4], through proteolytic cleavage—reviewed in [5].

In healthy individuals, blood ACE levels remain very stable during an individual’s lifetime [6], whereas in sarcoidosis and Gaucher’s disease, blood ACE is significantly increased—reviewed in [7]. Elevated ACE expression in tissues (which is generally reflected by blood ACE levels) is associated with an increased risk of cardiovascular diseases [1,8]. Whereas decreased ACE expression (accompanied by decreased blood ACE) is becoming a new risk factor for Alzheimer’s disease [9,10,11,12]. Therefore, correct (accurate) measurement of blood ACE (as a measure (read-out) of tissue ACE expression) has become clinically important for several diseases, especially in an era of precision (personalized) medicine [12,13,14].

In this study, we presented comprehensive ACE phenotyping in pairs of human tissues and serum from 50 individual male and female patients. Such ACE phenotyping of lung tissues (and corresponding sera samples) allowed us to identify patients with a unique ACE phenotype that shed new light on the mechanism of ACE inhibition by endogenous ACE inhibitor—human serum albumin [15]. Besides, simultaneous analysis of ACE levels in the lung and serum of each individual allowed us to establish an algorithm for the identification of still unknown ACE secretase.

Recently we demonstrated dramatic differences in urinary ACE activity precipitation (i.e., local ACE conformation) between males and females with several mAbs having overlapping epitopes, that include glycosylation sites N45/N117 on the N domain of ACE. Likely, this is explained by differences in the extent of sialylation of this particular glycosylation site in males and females. Moreover, we showed that the extent of sialylation, reflected by the 2H9/2D1 binding ratio was characteristic of ACE from different pooled human tissues (lung, heart and lymph nodes) and serum [16].

And, finally, we demonstrated that even serum ACE retains some sex differences in sialylation of the certain glycosylation site—N45, as evidenced by the calculation of the 2H9/2D1 binding ratio.

Thus, ACE phenotyping of human tissue and blood samples is a promising new approach with potential clinical significance to advance precision medicine screening techniques.

## 2. Methods

### 2.1. Chemicals

ACE substrates, benzyloxycarbonyl-l-phenylalanyl-l-histidyl-l-leucine (Z-Phe-His-Leu)—Cat. # 4000599) and hippuryl-l-histidyl-l-leucine (Hip-His-Leu)—Cat. # H1635) were purchased from Bachem Bioscience Inc. (Allentown, PA, USA) and Sigma (St. Louis, MO, USA), respectively. Other reagents (unless otherwise indicated) were obtained from Sigma-Aldrich (St. Louis, MO, USA). AM-15 and AM-4 ultrafiltration membranes (cut-off Mr 3000 and 100,000, respectively) were from Merck Millipore Ltd. (Cork, Ireland), and dialysis cassettes (cut-off 10,000) were from ThermoScientific (Rockford, IL, USA).

### 2.2. Antibodies

Antibodies used in this study include a set of 25 mAbs to human ACE, recognizing native conformation of the N and C domains of human ACE [17,18].

### 2.3. Study Participants

The collection of human samples used in this study was approved by the Ethics Committee of the University of Debrecen (Hungary) as described in detail previously [19]. All corresponding procedures were carried out in accordance with institutional guidelines and the Code of Ethics of the World Medical Association (Declaration of Helsinki). All volunteers and patients provided written informed consent to have different human tissue samples for ACE characterization. Tissue (lung, heart and lymph nodes) processing for further determination of ACE activity and immunoreactive ACE protein was performed as described in detail previously [19].

### 2.4. ACE Activity Assay

ACE activity in serum/plasma was measured using a fluorimetric assay with two ACE substrates, 2 mM Z-Phe-His-Leu or 5 mM Hip-His-Leu [20]. Calculation of ZPHL/HHL ratio [21] was performed by dividing the fluorescence of the reaction product produced by the ACE sample with ZPHL as a substrate to that with HHL. In some experiments, ACE activity measurement was performed with fluorogenic substrate Abz-FRK(Dnp)-P-OH and ACE protein level quantification—using an ELISA kit from the R&D System (Minneapolis, MN, USA), as described in detail previously [19].

### 2.5. Immunological Characterization of the Blood ACE

Microtiter (96-well) plates (Corning, NY, USA) were coated with anti-ACE mAbs via goat anti-mouse IgG (Invitrogen, Rockford, IL, USA) bridge and incubated with plasma/serum/lung ACE samples. After washing unbound ACE, the level of ACE immunoreactive protein was quantified as described previously using the strong mAb 9B9 [14,20]. Conformational fingerprinting of ACE was performed as described using a set of mAbs to different epitopes of ACE [17].

### 2.6. Whole Exome Sequencing

Genomic DNA was obtained from the blood clot of patient S13 (which serum and lung homogenate samples were analyzed under the name LMS1 and LM1, respectively) using a QIAamp DNA Mini Kit (Qiagen, Valencia, CA, USA). Whole exome sequencing (WES) and bioinformatic analysis of the sequencing data of DNA from 9 patients was carried out by Novogene (Sacramento, CA, USA).

### 2.7. Computational Analysis of the Models of ACE Interaction with Human Serum Albumin (HSA) and CCL18

The coordinates of X-ray structures of ligand-free and fatty acid-bound albumin, PDB:1E78 [22] and PDB:1E7H [23], respectively, were downloaded from the PDB. The models were aligned and rendered in PYMOL [24]. The dimer model of somatic ACE was prepared as described previously [16,25]. Subdomains 1 and 2 in the N domain of ACE were defined as described previously [26]. In the C domain, subdomain 1 contains residues 616–697, 876–1014,1118–1159 and subdomain 2—698–876,1015–1117,1160–1203. The 2D1 epitope is marked with a black oval.

PDB structures for ACE (7Q3Y) and for CCL18 (4MHE) were used for docking of CCL18 to somatic two-domain ACE.

### 2.8. Statistical Analysis

Values of ACE activity with different substrates for each individual, as well as other parameters characterizing the ACE phenotype, were means ± SD from 2–5 independent experiments (depending on the individual) with triplicates for each individual sample. Significance was analyzed using the Mann–Whitney test.

## 3. Results and Discussion

### 3.1. ACE Phenotyping in Human Lung Homogenates and Corresponding Sera Samples

ACE phenotyping was performed in nine male and nine female lung homogenates, diluted 1/20 in PBS, with two synthetic substrates (ZPHL and HHL). Figure 1A demonstrated ACE activity with short substrate ZPHL, while Figure 1B (ACE activity precipitated by mAb 9B9) showed immunoreactivity of individual ACEs in the lung homogenates. Both methods showed a high correlation in the quantification of ACE levels—r = 0.805 with mAb 9B9 (0.742, 0.674 and 0.782 for mAbs 1G12, 2H9 and 2D1, respectively). ACE activity and amount of ACE immunoreactive protein (with mAb 9B9) was unusually low in sample 1 (Lung Male 1—Figure 1A,B—blue boxed).

Naturally, lung tissue samples are not homogenous between individuals (with 5–20 mg of tissue per sample) as sera samples. If there is a larger vessel or fibrous scar, for example, it is making a large effect on ACE levels in a particular sample. Therefore, tissue ACE measurement is less reflective of ACE levels in a given individual than serum ACE measurement. If sample LM1 is excluded, variation of ACE levels in this cohort of 17 patients (Figure 1A,B) confirmed 3–4-fold individual variations in ACE levels in a bigger population—using sera samples [6,14,20].

In apparently normal individuals, blood ACE activities/levels vary from 50 to 150% of the mean—3-fold [14] and even bigger—17-fold in lung tissues [19]. ACE activity/levels in the lung homogenate of patient LM1 (Lung Male 1) is far less than 50% of the mean. The first suggestion (from Figure 1A) could be that the lung homogenate from this patient contains ACE inhibitor and medical history confirmed it—patient S13 (LM1) was prescribed with ACE inhibitor. However, there is some doubt about the presence of an exogenous ACE inhibitor in patient LM1 as the main reason for very low lung ACE activity in this patient. Lung homogenates were prepared with a 1:9 weight/buffer ratio and then 1/20 dilution of lung homogenate was taken for ACE activity, i.e., 1/200 was the final dilution of lung homogenate samples and nevertheless an apparent ACE activity in LM1 was just 28% from mean value. Thus, the concentration of putative ACE inhibitors in the lung of LM1 should be enormous.

Determination of ACE activity precipitated from this LM1 lung homogenate by four different mAbs (which include the step of washing out any exogenous ACE inhibitors bound to ACE on the bottom of the plate via mAbs and goat-anti-mouse bridge [20,27]) demonstrated that the apparent amount of catalytically active and immunoreactive ACE protein in the lung from patient LM1 is really low—much less than 50%—as shown by blue bars boxed in Figure 1B with mAb 9B9 and confirmed with three other mAbs (not shown).

Nevertheless, we calculated one parameter that allowed us to detect ACE inhibitors in serum or tissue samples—the ZPHL/HHL ratio [21]. Based on the increase of this parameter (Figure 1C), we can suspect that the lung homogenates LM1, LM6, LF3, LF4 and LF9 (orange bars) may contain ACE inhibitors.

Another parameter, 1G12/9B9 binding ratio, is applicable for plasma/serum sources of ACE and allows the detection of ACE inhibitors in the blood [27] because it is based on the ability of ACE inhibitors to induce dissociation of blood bilirubin from the binding site on the N domain of blood ACE [28]. Nevertheless, we calculated it also for lung homogenates and the high 1G12/9B9 binding ratio in patient LF9 (Figure 1D) may indicate the presence of ACE inhibitors in this patient. Interestingly, this parameter was not increased in the lung homogenate of patient LM1.

Finally, we calculated the 2H9/2D1 binding ratio for ACE in the lung homogenates from male and female patients. Previously, we demonstrated that this parameter, which likely reflects the extent of sialylation of glycosylation site N45 (and maybe N117) in the N domain of ACE, was several folds higher in male urinary ACE than in female urinary ACE [16]. The fact that this parameter was dramatically higher in the lung ACE of patient LM1 (more than 4-fold higher than in control lung homogenate) likely reflects much higher sialylation of this glycosylation site (N45/N117) in this patient and may indicate in favor of ACE mutation in patient LM1. If to exclude ACE from patient LM1 from the calculation of the mean value of this parameter for males and for females (because of putative ACE mutation in patient LM1), the 2H9/2D1 binding ratio for lung ACEs from males tends to be higher (+15%) than in females (albeit did not reach statistical significance). ACE activity itself did not show previously sex differences in human lung and heart tissues [19], or in human plasma/serum [29,30].

We performed similar ACE phenotyping on sera samples from these 18 patients (Figure 2). As with lung homogenates, serum ACE activity correlated well with ACE immunoreactive protein in the blood (for mAb 9B9 r = 0.793). ACE activity in the serum of patient LM1 was also very low (only 1/3 of the mean value in population, used as a control—Figure 2A)—as in lung homogenate.

However, serum ACE activity precipitated from patient LM1 by mAb 9B9 (Figure 2B) and by other mAbs (not shown), i.e., the amount of immunoreactive ACE protein, dramatically increased and became the same as in control. It may indicate that mAbs, immobilized on the bottom of the plate, did not work as chaperones for lung ACE from patient LM1, but in the serum of this patient, in the presence of 10,000 competing plasma proteins, mAbs started to work as chaperones, restoring native conformation of ACE in the serum of patient LM1. An alternative hypothesis is that putative and unknown ACE-binding protein in patient LM1 binds tightly to lung ACE, but not so tightly to blood ACE, and after precipitation by mAb 9B9 and extensive washing, it dissociates from blood ACE.

ACE phenotyping in 18 sera samples confirmed the suggestion (made using lung homogenates) that patients LM1, LM5 and LF5 contain ACE inhibitors in their tissues at the time of blood and tissue sampling (Figure 2C,D). And finally, the 2H9/2D1 binding ratio was also higher for male serum ACE (+15%) than for female serum ACE (but also did not reach statistical significance).

### 3.2. Further Analysis of ACE Outlier (from Patient LM1)

Since the discovery of PCSK9’s role in lipid metabolism, a new strategy for studying the proteins of interest has emerged: the deep analysis of phenotype outliers in combination with genetic analysis (whole exome or whole genome sequencing). This method has become very effective, because it does not only shed light on the mechanism of exceptional drug response (especially important for oncology targets) but is also crucial for precision medicine and sometimes leads to unpredicted discoveries of new functions of the studied proteins [31,32].

We applied this approach—analysis of ACE phenotype outliers, which could be considered as “individuals served as experiments of nature” [33]—already quite some time ago, and identified several new aspects of ACE biology. Analysis of ACE in patients with highly elevated blood ACE levels allowed us to discover the contribution of ACE dimerization in the regulation of ACE shedding [13,25], and the effect of lysozyme and bilirubin on ACE conformation, and again, on ACE shedding [28].

Thus, it was natural that we wanted to study the reason deeper for the unusual ACE phenotype of ACE in patient S13 (which lung and serum samples under the name LM1 we analyzed) [Figure 1 and Figure 2]. We also found that blood and lung ACE activity in patient S13 determined with longer substrate Abz-FRK (Dnp)P-OH was only 8% from control in comparison to 30% for shorter substrates ZPHL or HHL, which also fit with the hypothesis that patient S13 may have ACE mutation near active site entrance or groove (similar to what we identified earlier [34]).

In order to localize this putative ACE mutation, we performed conformational fingerprinting of lung ACE [17]—precipitation of ACE activity from lung homogenate of sample LM1 with 20 mAbs to different epitopes on the N and C domains of ACE [18]—and compared it with ACE activity precipitation from control lung homogenate (Figure 3A). The pattern of precipitation of ACE activity by a set of mAbs to ACE allowed us to predict the localization of the putative ACE mutation in patient S13 in the N domain, in the overlapping region of several epitopes: at first for mAb 2D1 and also for mAbs 6C8/6H6.

From this experiment, we also calculated the 2H9/2D1 binding ratio, which is the marker of sialylation of the particular glycosylation sites in the N domain of ACE—N45 and perhaps N117 [16]. This parameter was dramatically increased for ACE in the lung of S13—12.5 (414.6% from the mean (3.0)—for control lung homogenates)—but did not differ for serum ACE (78.7% of the mean (12.8)—because the absolute value for control ACE in the serum is already dramatically increased in comparison to lung ACE. Because 2D1 binds poorly to sialylated ACE, as in serum [16], we can suggest that mutant ACE in the lung of the S13 patient is already highly sialylated and thus, we can exclude at least asparagine in position 45 and serine/threonine in position 47 from candidates for putative ACE mutation (because glycosylation site in this position should be preserved).

This putative point mutation likely may also change conformation in the region on the C domain (overlapping epitopes of mAbs 1E10/4E3/2H9) which is in close proximity to the N domain [25]. The specific pattern-dramatic drop of 2D1 binding and absence of changes in 5F1 binding, which is highly overlapped with epitope for 2D1 (Figure S3 in [18]), even suggested that K107 could be mutated in patient S13 (see Figure 3C,D in [18]).

The conformational fingerprint of ACE (Figure 3) and ACE phenotype in patient S13 (Figure 1 and Figure 2) as well as the effect of albumin on mAbs binding to ACE [35] prompted us to hypothesize that this putative ACE mutation may change the binding of albumin to ACE and thus, may change the effect of albumin on ACE inhibition [15]. Figure 4A,B demonstrated dramatic differences in lung ACE inhibition in patient S13 and in control lung homogenates by human serum albumin (HSA), especially by albumin preparation depleted of free fatty acids. A comparison of ligand-free and fatty acid-bound albumin structures (Figure 4C) shows pronounced overall differences in their conformations and provides a rationale for the differences in ACE inhibition.

The docked model of ACE-albumin binding (Figure 5) exemplifies a possible mechanism of albumin-mediated inhibition of ACE. HSA docked to ACE interacts with both N and C domains. HSA forms large contact with both the subdomains of the C domain that enclose the catalytic site. In the C-domain, HSA is expected to interfere with the catalytic machinery and intra-domain cooperativity either by impeding the binding of the substrate or by interfering with the motion of ACE subdomains 1 and 2 relative to each other (see “Jaws” in Figure 5). Interaction of HSA with N domains has a smaller footprint on the N domain. HSA interacts mostly with subdomain 2 in the vicinity of epitope 2D1 which can also result in impeded mobility of the N subdomains 1 and 2 due to subdomain 2 anchored to HSA. Considering the position of HSA between the N and C domains at the linker region, we also cannot exclude the involvement of HSA in blocking the inter-domain and dimerization cooperativity. In all cases, such interference is likely to affect the efficiency of the ACE catalytic mechanism [38,39,40,41,42].

Therefore, to identify putative ACE mutation in patient S13 as a reason for the unusual ACE phenotype in this patient, we performed whole exome sequencing of the DNA from patient S13 (Appendix A). To our surprise, no missense or damaging ACE mutations were identified in this patient (Appendix A). The next hypothesis we considered was that this subject may have a mutation in one of the known ACE binding proteins: chaperone BiP (GRP78), ribophorin 1, protein kinase C [43], albumin [15], lysozyme [28]. In the case of such mutations, altered binding of these proteins to ACE may change the local conformation of ACE, leading to changes in the ZPHL/HHL hydrolysis ratio. However, sequencing of S13 genomic DNA did not detect any loss-of-function (LoF) mutations in any of these known ACE-binding proteins.

What we had at that moment was the following: (1) ACE activity in serum and in the lung is very low with short ACE substrates and even lower, with longer ACE substrates; (2) immobilization of ACE by mAbs (with a decrease in ACE flexibility) restore the complete catalytic activity of serum ACE from patient S13 and to some extent for lung ACE; (3) the binding of some mAbs on the N domain (and in adjacent region on the C domain) where albumin binds to ACE was changed; (4) albumin depleted of free fatty acids lost its ability to inhibit ACE activity in the lung of patient S13; (5) there is no ACE mutation in patient S13 that could explain this unusual ACE phenotype.

Such properties allowed us to propose an alternative hypothesis for the unusual ACE phenotype in this patient. We suggest that patient S13 has dramatically increased concentration in the lung of unknown ACE binding protein that binds to the N domain in the region of the epitope for mAb 2D1 (which is close to the epitope for mAb 1E10 on the C domain of ACE) and changed its catalytic properties. We found previously analogous ACE-binding protein (with MW between 10 and 30 kD) in the normal spleen tissue, which disappeared in the spleen of patients with Gaucher disease [44]. In our case, it seems that this yet unidentified ACE-binding protein binds tightly to lung and blood ACE in solution, but dissociates from serum ACE if this ACE is immobilized by mAbs (Figure 2B).

Testing of an additional eight sera samples with limited conformational fingerprinting (with six mAbs to ACE) demonstrated that another patient (TX-18) showed similar (as in patient S13) changes in the local conformation of ACE (Figure 3B) in the regions of the epitopes of mAbs 2D1/5F1 on the N domain and epitope for mAb 1E10 on the C domain, i.e., on the borders of the cleft (pocket) between N and C domains of ACE (Figure S10 in [28]). Therefore, changes in the local conformation of ACE near the cleft between N and C domains could be not unique for patient S13 and may have pathophysiological (clinical) significance, because carriers of this ACE phenotype will have less inhibition of ACE by natural endogenous ACE inhibitor—albumin [15], and thus, due to high concentration of Angiotensin II, be susceptible for different cardiovascular complications [45].

To find a relation between our case with patient S13 and spleen ACE and Gaucher diseases, we titrated homogenates of normal lung and spleen and spleen from patient with Gaucher disease [44] with albumin and demonstrated that native human serum albumin preparation did not inhibit ACE activity in the spleen homogenates from four patients with Gaucher diseases, while did so in the spleen and lung homogenates of unrelated patients (Figure 6A). Preferential inhibition of ACE activity with substrate Hip-His-Leu, which hydrolyzed much faster by the C domain active center of ACE [21,46], confirmed previous observations that the C domain active center is a primary target for inhibition by albumin [15]. The increased ability of albumin to inhibit ACE activity in the spleen in comparison to the lung (Figure 6A–C) could be related to better accessibility for albumin to spleen ACE surface in the region of the epitopes for mAbs 1E10 and 2H9 on the C domain and less accessibility to the region of epitopes for mAbs 6C8 and 5F1 on the N domain (red box in Figure 6D), especially because the availability of these ACE surfaces in Gaucher spleen for these mAbs was inversed (green box in Figure 6E) and fitting well with the complete inability of albumin to inhibit ACE in Gaucher spleen (Figure 6A–C).

Previously, we proposed that small proteins (10–30 kD) like lysozyme, beta2-microglobulin, cystatin, transthyretin, and serum amyloid protein A may bind to ACE exactly in the cleft between N and C domains and identified that at least lysozyme binds to ACE [28] in this cleft. We even proposed that not only lysozyme, but likely other proteins of such size can also bind to ACE—just because they fit with this cleft volume—see Figure S10 in [28]. A reliable suggestion to explain these results is that the failed ability of albumin to inhibit ACE in the lung homogenate of patient S13 and in the spleen homogenate of patients with Gaucher diseases could be due to the appearance in the cleft of an unusually small protein (10–30 kD), occupying this cleft and competing with albumin for the binding site on ACE (Figure 6F). Moreover, bearing in mind that conformational fingerprint (i.e., pattern of mAbs binding to these regions on the C terminal end of the N domain (mAbs 6H6/2D1/6C8/5F1) and on the N terminal end of the C domain (mAbs 1E10/2H9/3C10/4E3)) differ in the case of lung ACE of patient S13 (Figure 3) and spleen ACE of patients with Gaucher diseases (Figure 6E), the nature of these ACE binding proteins could be even different. Identification of ACE-binding protein in the normal spleen, which disappears in the spleen of Gaucher disease [44] may help with the identification of the mutation of unknown ACE modifier (ACE-binding protein) in patient S13.

Whole exome sequencing of genomic DNA from patient S13 identified 10,000+ changes in its exome in comparison to the reference sequence (Appendix A), which includes 1381 damaging missense mutations (according to PolyPhen-2 [47], HVAR score) and 641 exonic nonsense/indels mutations (Appendix A). Among numerous mutations in patient S13, at least two mutations attracted our attention as possibly causal for specific ACE phenotype in this patient: mutation rs550365194 in gene *SMPD1* (ASM) and mutation rs1136747-V75A in gene *SAA1* (serum amyloid protein A). In order to find mutations responsible for specific ACE phenotype in patient S13, we subtract from damaging mutations in patient S13 (Appendix A) all random damaging mutations that we found in 16 subjects, for which whole exome sequencing was performed during the last two years (Appendix A), but these subjects did not demonstrate this specific ACE phenotype [14,16] as in patient S13. Such subtraction (which we already performed for another case [14]) eliminated about 1700 random gene variants in patient S13 (including mutations that were suspicious for us (*SAA1* and *SMPD1*)) and selected 200 + mutations—possible candidates to be causal for such ACE phenotype (Appendix A). It is highly probable that the story with patient S13 ACE phenotype is not a case of monogenic syndrome—when a specific phenotype is due to one causal mutation in one gene—rather, it may represent the most common scenario—when clinical symptoms/syndromes in a particular patient is a result of the **combination** of several heterozygous mutations in several genes.

In the case of patient S13 it could be a combination of

*SAA1*—rs1136747 (V75A);*ADAMTSL4*—rs199599791 (R87P);*NOD2*—rs2066844 (R702W);*SMPD1*—rs550365194 (V36-V39 del).

Disruption in NOD2 signaling has been associated with impaired host defense and associated with Crohn’s disease and Blau syndrome [48], The mutations in the *SMPD1* gene are causal for Niemann–Pick disease in homozygous format [49].

Despite the fact, that some of these mutations are also present in some of the 16 patients with WES, but without specific ACE phenotype (Appendix A), such **combination** is present only in S13, and, as a result, such combination of different small proteins, associated with these mutations, may accumulate in the pocket (cleft) between N and C domains of ACE and prevent binding of albumin to ACE in patient S13. Therefore, we decided to test the possible involvement of two suspicious proteins, SAA1 and CCL18, in specific ACE phenotypes in patient S13.

The protein coded by gene *SAA1* is serum amyloid protein A, an acute phase protein, whose concentration in the blood increases up to 1000-fold in cancer, as a response to infection, injury and inflammation [50,51]. The mutation found in patient S13—(rs1136747-V75A) looks like a gain-of-function *SAA1* mutation, because SAA1 levels in the blood of carriers of this mutation increased 1.5-fold [52]. Besides, patients with lung cancer had an 18-fold elevation of SAA1 in the blood [53] and patients S13 had lung adenocarcinoma, grade 2. Therefore, due to both reasons, we may expect an increase of SAA1 protein in the blood and/or tissues of S13 (and maybe TX18). Measurement of SAA1 in S13 plasma and tissues) versus proper controls—control serum and/or control lung (or control heart) homogenates did not demonstrate an increase in SAA1 in patients S13 and SAA1 did not increase ZPHL/HHL ratio on two types of soluble ACE—lung ACE homogenate or seminal fluid ACE. Therefore, we can exclude changes in SAA1 content in patient S13 as a reason for the unusual ACE phenotype.

The protein coded by gene *SMPD1* (Uniprot P17405 https://www.uniprot.org/uniprotkb/P17405/entry (accessed on 1 January 2024) converts sphingomyelin to ceramide. The *SMPD1* mutation (rs550365194) found in patient S13 is an inframe deletion of 3 amino acids—p.Val36_Leu39del—and thus, likely damaging. The mutations in this gene are associated with Niemann–Pick disease and thus, a high concentration of chitotriosidase and CCL18/PARC in the blood of this patient with such disease is quite probable [54]. Unfortunately, we do not have any more blood samples from this patient S13 for analysis for chemokine CCL18). In order to test the possibility that chemokine CCL18/PARC, which blood concentration dramatically increased in patients with Gaucher disease and other lysosomal storage disorders [49,54,55], we incubated lung and spleen homogenates from unrelated patients and homogenates from patients with Gaucher disease with CCL18/PARC; we found that CCL18 did not change the ZPHL/HHL ratio for lung ACE homogenate, and for spleen homogenate from patients with Gaucher disease who had ZPHL/HHL ratio closed to this parameter for lung ACE [44], the ratio decreased significantly for normal spleen homogenate, which normally had twice higher initial ZPHL/HHL ratio (Appendix A). Thus, likely CCL18 could be considered a novel ACE-binding protein.

Therefore, it seems the hypothesis that the ACE phenotype in the S13 patient (which is similar to the phenotype of ACE in Gaucher spleen) could be explained by the binding of the excess of CCL18 (due to Niemann–Pick) to ACE in the S13 patient became more probable. The model showing the docking of CCL18 to the cleft between N and C domains in the cryo-EM structure of two-domain somatic ACE [42] is shown in Figure 7.

We suggest that some similarities in ACE phenotype in the lung ACE from patient S13 and in the Gaucher spleen ACE could be explained by some similarities in local ACE glycosylation (N45/N117 in the N domain and in N666/N685 in the C domain) in these tissues.

### 3.3. Sex Differences in ACE Sialylation in Different Tissues

We also performed ACE phenotyping in a set of six samples of heart tissue and corresponding sera samples in males and females and in a set of ten samples of lymph nodes and corresponding sera samples. We found only one outlier in ACE phenotype in these samples: ACE in the heart tissue homogenate and serum from patient HF2 (heart female #2) has significantly decreased ZPHL/HHL ratio (approximately 60% of control value, *p* = 0.005) which may indicate that this patient may have ACE mutation near active site, positioning substrate binding (analogous to ACE mutation S333W that we identified previously [34].

We also calculated the 2H9/2D1 binding ratio for these tissues and sera samples and found that this ratio was practically similar for ACE in male and female heart and lymph node tissue homogenates—Appendix A), which indicates of similar extent of sialylation of at least two glycosylation sites (N45 and N117, which localized in the epitope for mAb 2D1). Of note, the 2H9/2D1 ratio for ACE in lymph nodes of patients with sarcoidosis was much lower than in ACE in lymph nodes of unrelated patients (red bars in Appendix A), which is rather a reflection of too low ACE activity in the homogenates of lymph nodes of unrelated patients. Appendix A demonstrated that the 2H9/2D1 ratio for a bigger cohort of sera samples weakly (just +46%) but statistically significantly higher for sera samples from males than from females (which confirmed our previous results obtained on less representative population [16]. The difference in this ratio between males and females is small, in comparison to urinary ACE (+700% in [16]) and could not be used for diagnostic purposes, but indicates existing differences in ACE sialylation between males and females in at least some sites of glycosylation in male and female ACEs in tissues.

## 4. Conclusions

During ACE phenotyping in human tissues and serum, we identified patient S13 with a unique ACE phenotype—related to the ability to inhibit ACE activity by albumin. Whole exome sequencing of the genomic DNA of this patient did not reveal ACE mutation as a reason for such phenotype.

An alternative suggestion was that patient S13 has dramatically increased concentration in the lung of an unknown ACE-binding protein that binds to the N domain in the region of the epitope for mAb 2D1 (which is close to the epitope for mAb 1E10 on the C domain of ACE) and changed its catalytic properties. We found previously another ACE-binding protein (with MW between 10 and 30 kD) in the normal spleen tissue, which disappeared in the spleen of patients with Gaucher disease [44]. In our case, it seems that this yet unidentified ACE-binding protein binds tightly to lung and blood ACE in solution, but dissociates from serum ACE if this ACE is immobilized by mAbs (Figure 2B).

Our data allowed us to hypothesize that such ACE-binding protein is chemokine CCL18 and ACE phenotype in the S13 patient (which is similar to the phenotype of ACE in Gaucher spleen) could be explained by binding of the excess of CCL18 (due to Niemann–Pick disease phenotype in this patient) to ACE in the S13 patient, which prevents anti-catalytic action of albumin towards ACE.

Therefore, ACE phenotyping is a promising new approach for the identification of ACE phenotype outliers with potential clinical significance, making it useful for screening in a personalized medicine approach.

## Figures and Tables

**Figure 1 biomedicines-12-00940-f001:**
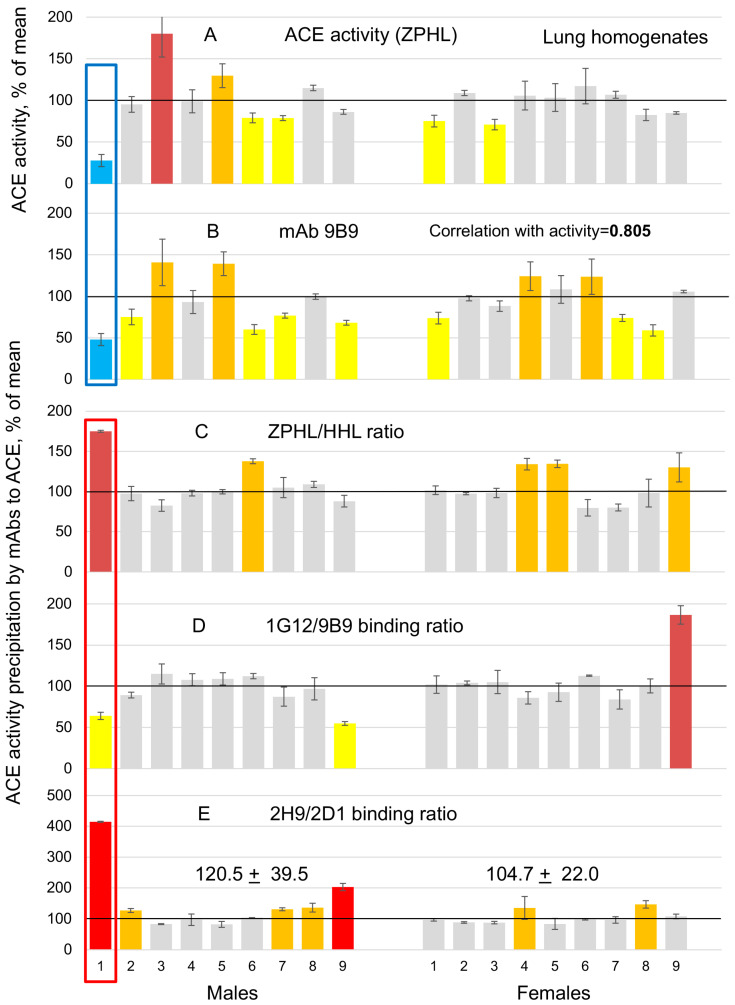
ACE phenotyping in human lung homogenates. ACE activity in 18 samples of human homogenates (1:9 weight/volume ratio, diluted 1/20 in PBS, 9 males and 9 females) was quantified using a spectrofluorometric assay with Z-Phe-His-Leu (**A**) and Hip-His-Leu (not shown) as substrates. ACE activity from human lung homogenates was precipitated by mAb 9B9 (**B**) and by mAbs 1G12, 2H9 and 2D1 (not shown). (**C**) Ratio of the rate of hydrolysis of the two substrates (ZPHL/HHL ratio) in the tested samples. (**D**) 1G12/9B9 binding ratio. (**E**) 2H9/2D1 binding ratio. Data expressed as % of individual ACE activity in solution (**A**) or precipitated by mAbs (**B**–**E**) from mean values for all samples. Bars with significant changes in % of control ACE activity are colored as follows: increase more than 20%—with orange, >50%—with brown, more than 2-fold—with red, decrease more than 20%—yellow, more than 50%—light blue. Grey bars represent data within normal values between 80 and 120% of control. Mean values (+SD) from 2–5 experiments (each made in triplicates).

**Figure 2 biomedicines-12-00940-f002:**
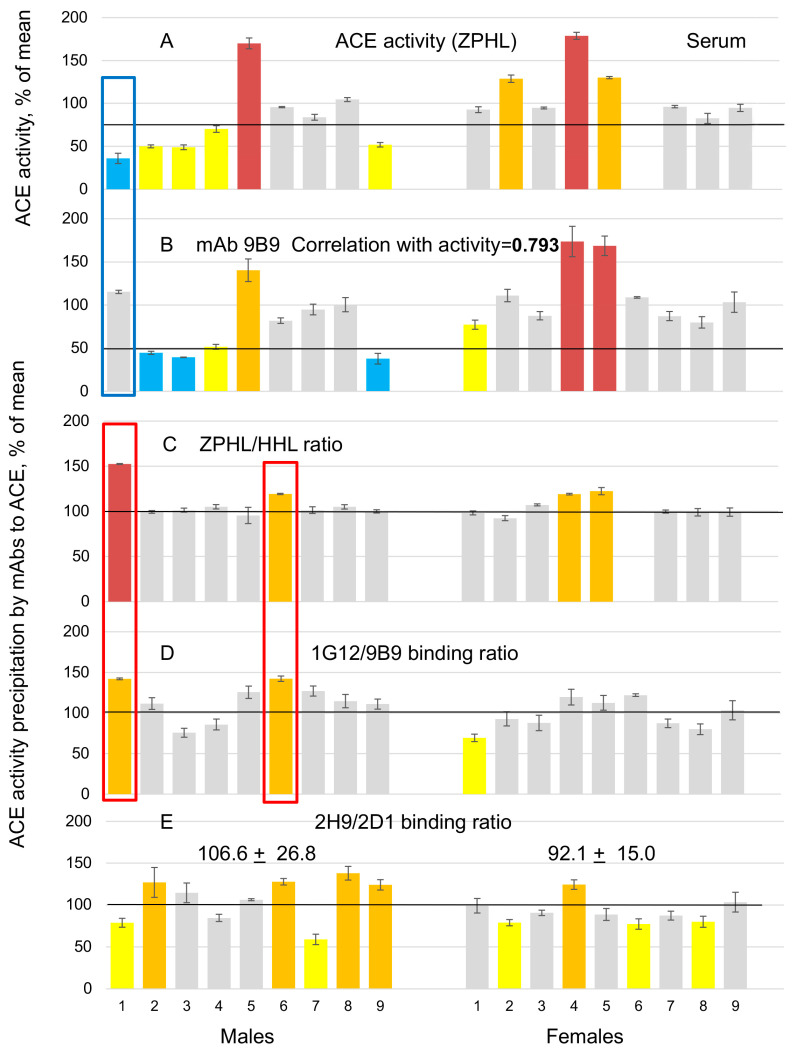
ACE phenotyping in human serum. In total, 18 samples, diluted 1/15 in PBS, which are the pairs from the same patients as tested lung tissues, were performed exactly as in Figure 1.

**Figure 3 biomedicines-12-00940-f003:**
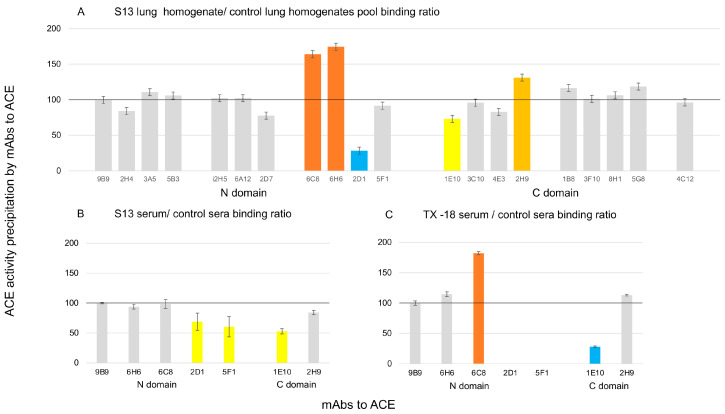
Conformational fingerprinting of ACE from patient S13. ACE activity was precipitated from lung homogenates of patient S13 (diluted 1/20) versus pooled lung homogenates of patients 2 to 9 (**A**) by 20 mAbs to different epitopes of human somatic ACE. Analogous conformational fingerprinting of blood ACE from patient S13 (**B**) and patient TX-18 (**C**) was performed with 7 mAbs to ACE. Data presented as ratio (%) of ACE activity precipitation from lung (**A**) to that from control lungs or serum (**B**,**C**) or that from control serum. Mean values (+SD) from 2 experiments (each made in triplicates). Bars are colored as in Figure 1.

**Figure 4 biomedicines-12-00940-f004:**
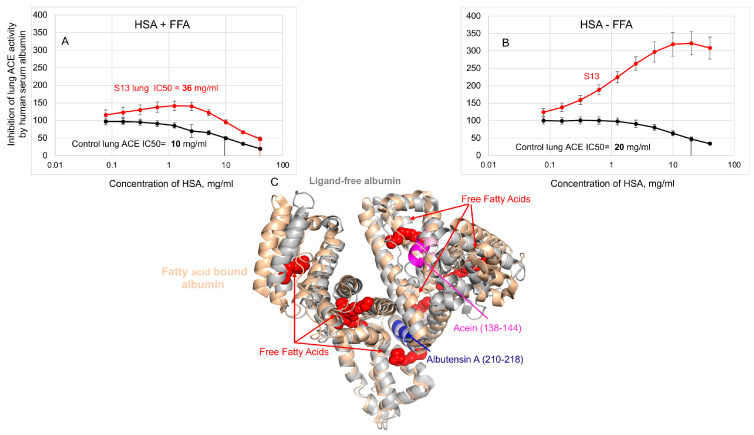
Effect of human serum albumin on lung ACE activity from patient S13. (**A**,**B**) Lung homogenate of patient S13 (red) and control pooled lung homogenate (black) were titrated with different concentrations of human serum albumin (HSA)—free fatty acid-bound albumin (HAS + FFA) and FFA-depleted HSA-preparation (HSA-FFA). After equilibration, ACE activity was determined with fluorometric assay with fluorogenic substrate (Abz-FRK(Dnp)P-OH), as described earlier [19]. Data expressed as a % of residual activity. (**C**) Superposition of ligand-free (gray, PDB: 1E78) and fatty acid-bound (wheat, PDB:1E7H) human albumin. Fatty acids are rendered as van der Waals spheres in red. The sequence 138-144, Acein-1 [36], is rendered in magenta and light pink colors in ligand-free and fatty acid-bound albumin, respectively. The sequence 210–218, Albutensin A [37], is rendered in blue in ligand-free and fatty acid-bound albumin.

**Figure 5 biomedicines-12-00940-f005:**
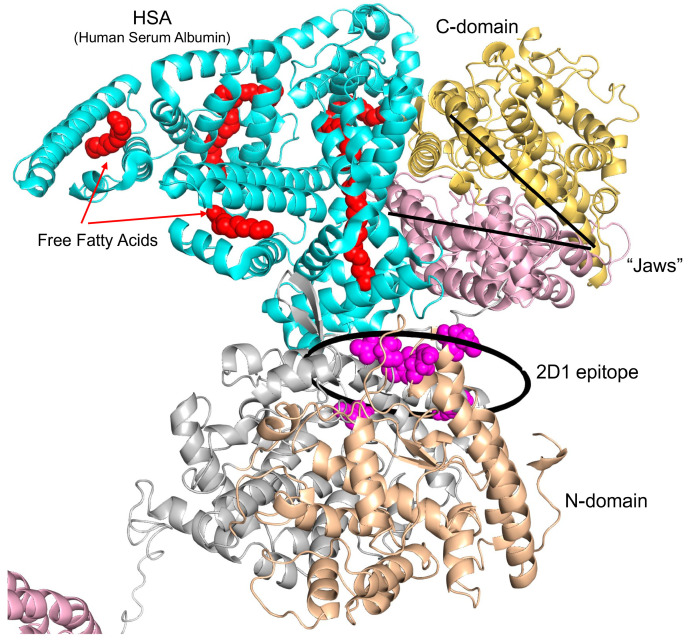
Albumin docking to ACE dimer. Human somatic ACE model presented as a dimer [25] was used for modeling human albumin docking to ACE. Docked structure of fatty acid-bound (cyan, PDB:1E7H) albumin to a dimer model of ACE [16]. Two molecules of albumin can bind to the ACE dimer, but only one albumin molecule is shown to show which ACE epitopes interact with albumin. Fatty acids are rendered as van der Waals spheres in red. The subdomains I and II of N domain of ACE are rendered with grey and wheat colors, respectively. The subdomains I and II of C domain of ACE are rendered with pink and yellow colors, respectively. Epitope for mAb 2D1 showed schematically as a circle with 600 A^2^.

**Figure 6 biomedicines-12-00940-f006:**
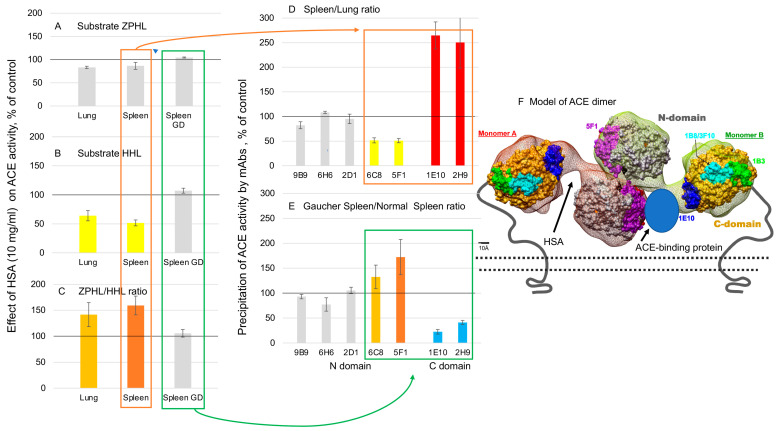
Inhibitory effect of human serum albumin (HSA) on ACE activity. Lung and spleen homogenates (1/20 dilution in PBS) were incubated with HSA at final concentration 10 mg/mL and residual ACE activity was determined fluorometrically with ZPHL (**A**) and HHL (**B**) as substrates. (**C**) Ratio of the rates of hydrolysis by two substrates. Data are mean ± SD from 3 independent experiments performed in triplicates. Three spleen tissues were from patients with Gaucher disease. ACE activity was precipitated from spleen and lung homogenates of unrelated patients (**D**) and spleen homogenates from patients with Gaucher disease (**E**). Data are presented as binding ratios for each mAb (mean ± SD from 3 independent experiments performed in triplicates). (**F**) Model of ACE dimer (adapted from [25]) where docking of HSA (from Figure 5) was shown by black arrow on one monomer and putative ACE binding protein preventing albumin binding was shown on another monomer in this ACE dimer.

**Figure 7 biomedicines-12-00940-f007:**
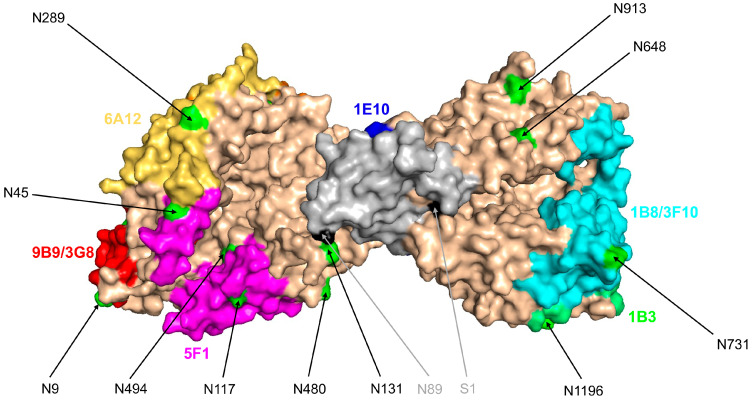
Model of CCL18 docking to ACE. A model of human somatic ACE (PDB 7Q3Y [42]) was used for modeling CCL18 docking to ACE. The surface ACE is colored in beige. CCL18 structure (PDB 4MHE) is shown in gray. The epitopes for mAbs on the N and C domains of ACE are shown in different colors. Asparagine (N) residues of the putative glycosylation sites are highlighted in lime green. The N and C terminal residues of CCL18 are highlighted in black.

## Data Availability

The original contributions presented in the study are included in the article/Appendix A and further inquiries can be directed to the corresponding authors.

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
