# Peer review of "ACE Phenotyping in Human Blood and Tissues: Revelation of ACE Outliers and Sex Differences in ACE Sialylation"

_biomedicines, 2024, doi:10.3390/biomedicines12050940_

Round 1
Reviewer 1 Report
Comments and Suggestions for Authors
The study investigates into characterizing Angiotensin-Converting Enzyme (ACE) phenotypes in patients with cardiovascular diseases through a novel method called ACE phenotyping. It examines ACE activity, immunoreactive protein levels, and conformation in tissue and serum samples from 50 patients, aiming to identify unique ACE phenotypes and gender differences in sialylation of ACE glycosylation sites. However, potential biases or confounding factors that may influence result interpretation should be considered.
The inclusion and exclusion criteria for patient selection must be explicitly defined to ensure the study's validity.
Although the study enrolls 50 patients, it only conducts whole exome sequencing from 8 patients. The rationale for selecting this subset from the larger sample pool should be clarified, likely relating to the focus on specific genetic variations or markers.
Differentiating from the authors' prior work, such as "Urinary ACE Phenotyping as a Research and Diagnostic Tool: Identification of Sex-Dependent ACE Immunoreactivity," this study likely extends the investigation into ACE phenotyping to tissue and serum samples, broadening the scope of understanding.
Limitations of the study, such as the small sample size for whole exome sequencing, must be acknowledged.
Regarding ACE phenotyping, there seems to be a discrepancy between the abstract and methodology sections regarding the number and types of samples subjected to analysis. This discrepancy needs clarification to ensure the accuracy of the study's methodology.
Overall, the methodology should provide clear details on the number and types of samples subjected to ACE phenotyping to avoid confusion and ensure transparency in the research process.
Comments on the Quality of English LanguageMinor improvement is needed
Reviewer 2 Report
Comments and Suggestions for Authors
I would like to recommend this manuscript for publication, but several small problems should be corrected:
1. There are too many keywords in the manuscript, are each one a keyword?
2. Please unify the interval punctuation in Keywords, is it a semicolon or a comma?
3. In Section 2.8 Statistical analysis, "means ± SD from 2-5 independent experiments (depending on individual)
with triplicates", how to obtain triplicates from 2 experiments
4. A western blot or PCR characterization is suggested to prove the ACE is used.
5. Author Contribution should be declared in the end of the manuscript.
6. Due to the use of human blood samples in the article, authorization
from the ethics committee is required.
7. Is the molecular structure simulation diagram in Fig. 4-Fig.7 calculated
by the authors themselves or cited from literature? If it is cited from literature,
relevant copyright permissions are required.
Round 2
Reviewer 1 Report
Comments and Suggestions for Authors
The authors have made substantial changes in several part of the paper and addressed the reviewers’ comments. This manuscript may be accepted for publication.
Author Response
Official permission from the original creators (PLOS One) of figure 6.
Link to PLOS One Policy:
https://journals.plos.org/plosone/s/licenses-and-copyright#loc-give-proper-attribution
and excerpt from their policy itself:
Reuse of PLOS Article Content
PLOS applies the Creative Commons Attribution 4.0 International (CC BY) license,
or other comparable licenses that allow free and unrestricted use, to articles and other works we publish. If you submit your paper for publication by PLOS,
you agree to have the CC BY license applied to your work.
If your institution or funder requires your work or materials to be published under a different license or dedicated to the public domain –
for example Creative Commons 1.0 Universal (CC0) or Open Governmental License –
this is permitted for those licenses where the terms are equivalent to or more permissive than CC BY.
PLOS requires that you as the author agree that anyone can reuse your article content in whole or part for any purpose, for free, even for commercial purposes.
These permitted uses include but are not limited to self-archiving by authors of submitted, accepted and published versions of their papers in institutional repositories.
Anyone may copy, redistribute, reuse, or modify the content as long as the author and original source are properly cited.
This facilitates freedom in reuse and also ensures that PLOS content can be mined without barriers for the needs of research.
We properly cited in the legend to Fig.6F:
- F. Model of ACE dimer (adapted from [25]) where docking of HSA (from Fig.5)
- Danilov, S.M.; Gordon, K.; Nesterovitch, A.B.; Luensdorf, H.; Chen, Z.; Castellon, M.; Popova, I.A.; Kalinin, S.;
Mendonca, E.; Petukhov, P.A.; Schwartz, D.E.; Minshall, R.D.; Sturrock, E.D.
Angiotensin I-converting enzyme mutation (Y465D) causes dramatic increase in blood ACE
via accelerated ACE shedding due to changes of ACE dimerization. PLoS One. 2011, 6, e25952.
